# Sulfated Hydrogels as Primary Intervertebral Disc Cell Culture Systems

**DOI:** 10.3390/gels10050330

**Published:** 2024-05-14

**Authors:** Paola Bermudez-Lekerika, Katherine B. Crump, Karin Wuertz-Kozak, Christine L. Le Maitre, Benjamin Gantenbein

**Affiliations:** 1Tissue Engineering for Orthopaedics and Mechanobiology, Bone & Joint Program, Department for BioMedical Research (DBMR), Medical Faculty, University of Bern, 3008 Bern, Switzerland; paola.bermudez@unibe.ch (P.B.-L.); katherine.crump@unibe.ch (K.B.C.); 2Graduate School for Cellular and Biomedical Sciences (GCB), University of Bern, 3012 Bern, Switzerland; 3Department of Biomedical Engineering, Rochester Institute of Technology, Rochester, NY 14623, USA; kwbme@rit.edu; 4Spine Center, Schön Klinik München Harlaching Academic Teaching Hospital, Spine Research Institute, Paracelsus Private Medical University Salzburg (Austria), 81547 Munich, Germany; 5Division of Clinical Medicine, School of Medicine and Population Health, University of Sheffield, Sheffield S10 2TN, UK; c.lemaitre@sheffield.ac.uk; 6Inselspital, Department of Orthopedic Surgery & Traumatology, Medical Faculty, University of Bern, 3010 Bern, Switzerland

**Keywords:** intervertebral disc, sulfation alginate, 3D culture, qPCR, cell viability

## Abstract

The negatively charged extracellular matrix plays a vital role in intervertebral disc tissues, providing specific cues for cell maintenance and tissue hydration. Unfortunately, suitable biomimetics for intervertebral disc regeneration are lacking. Here, sulfated alginate was investigated as a 3D culture material due to its similarity to the charged matrix of the intervertebral disc. Precursor solutions of standard alginate, or alginate with 0.1% or 0.2% degrees of sulfation, were mixed with primary human nucleus pulposus cells, cast, and cultured for 14 days. A 0.2% degree of sulfation resulted in significantly decreased cell density and viability after 7 days of culture. Furthermore, a sulfation-dependent decrease in DNA content and metabolic activity was evident after 14 days. Interestingly, no significant differences in cell density and viability were observed between surface and core regions for sulfated alginate, unlike in standard alginate, where the cell number was significantly higher in the core than in the surface region. Due to low cell numbers, phenotypic evaluation was not achieved in sulfated alginate biomaterial. Overall, standard alginate supported human NP cell growth and viability superior to sulfated alginate; however, future research on phenotypic properties is required to decipher the biological properties of sulfated alginate in intervertebral disc cells.

## 1. Introduction

Low back pain (LBP) is a widespread symptom experienced by people of all ages [1,2,3]. Several types of pain fall under the umbrella of the LBP spectrum and can simultaneously overlap, including nociceptive and neuropathic pain (radicular) as well as non-specific pain often associated with the amplification of pain in the central nervous system (CNS) [4].

Since 2015, LBP has remained the leading cause of Years Lived with Disability (YLD) globally, with approximately 619 million prevalent cases in 2020, which is predicted to rise to 800 million by 2050. Interestingly, a higher global prevalence is reported among females compared to males with a higher difference over 75 years of age group [5]. In terms of economic burden, the annual costs related to patient management with LBP exceeded USD 100 billion in the US [6], GBP 2.8 billion in the UK [7], AUD 4.8 billion in Australia [5], and EUR 2.6 billion in Switzerland [8], demonstrating an economic impact comparable to other prevalent diseases including cancer, autoimmune diseases, mental health and cardiovascular diseases [9].

According to the recent Global Burden of Disease risk factors study, almost 40% of years lived with disability (YLD) caused by LBP are attributed to mechanical and ergonomic factors (e.g., lifting, forceful movements, awkward postures), smoking and high BMI [10]. Nevertheless, although disc-related diseases have not been accurately classified as individual pain contributors [11], several clinical and imaging studies have associated back pain with the degeneration of the intervertebral disc (IVD) [12], with IVD degeneration considered a significant cause of LBP [13,14].

IVDs are located between two adjacent vertebral bodies and contribute to the weight bearing of the spine, withstanding compression loads, and providing spine flexibility [15,16]. An IVD consists of three primary tissues: the nucleus pulposus (NP) core, peripherally constrained by the annulus fibrosus (AF) region which are constrained cranially and caudally by the cartilage endplates (CEP) [17]. The IVD is the largest avascular organ in the human body, with blood vessels only present in the vertebrae boney end plates and outer AF region [18]. Nevertheless, nutrient, oxygen and waste transport are possible inside the core of the NP and inner AF due to the CEP’s key role as a diffusion intermediary from the peripheral vasculature [19]. Under physiological conditions, the central NP region contains high concentrations of negatively charged sulfated proteoglycans such as aggrecan molecules, which are responsible for the osmotic swelling of the disc. Similarly, the healthy AF is a highly organized fibrous structure with alternate collagen lamella fibers, which are orientated at ±30° consisting of predominately type I collagen in the outer AF [20]. During the degeneration of the IVD, cell phenotype changes with a decrease in the normal disc matrix and an increased degrading enzyme production, leading to the loss of proteoglycan in the NP and thus a decreasing water content [21], disc height [22] and promoting the generation of more fibrotic tissue [23]. The degeneration of the AF leads to disc bulging [24] as well as the accumulation of structural defects [25,26], together with increased fissures and the calcification of the CEP [27], leading to altered permeability [28]. Thus, as the IVD degenerates, structural and biochemical changes occur, leading to blood vessel and nerve ingrowth into the disc, which is thought to contribute to pain.

In the past decades, several novel biomaterials have been tested in in vitro studies to more closely mimic the native IVD extracellular matrix (ECM) with the aim to provide improved physiological mechanical support, cytokines and growth factors reservoir. Alginate has been extensively employed in IVD research due to its structural similarity to native ECM, low toxicity [29], high biocompatibility [30] and ability to be modified [31]. Nevertheless, although alginate is considered a gold-standard hydrogel in IVD and cartilage fields [32], it lacks a crucial feature which is present in IVD and cartilage ECM: the sulfonic groups of proteoglycans responsible for negatively charged microenvironments [33,34]. Native IVD ECM is mainly composed of proteoglycans, which consist of a core protein modified by linear sulfated glycosaminoglycans (GAGs), responsible for osmotically water attraction conferring mechanical stability and local biological environment modulation. These GAGs bear negative charges that are essential for developing biomimetic hydrogels [35,36]. The negative charges characteristic of sulfated hydrogels provide two beneficial effects: (1) chemical cues to maintain and induce chondrocytes/disc-like phenotype [37,38,39,40] and (2) the possibility of the constant release of positively charged drugs [41,42]. Studies investigating articular chondrocytes have shown that sulfated alginate carriers provide potential alternatives to non-sulfated alginate due to a higher cell viability, increased proteoglycan deposition, chondroprotective effects and sulfation dose-dependent cell proliferation, recognition and adhesion, as well as an increased fibroblast growth factor (FGF) retention in human or bovine articular chondrocytes [43,44]. Nevertheless, standard alginate culture has exhibited improvements compared to the sulfated alternative, including increased collagen type II deposition and a higher stiffness in bovine articular chondrocytes [43,45]. To date, their use has not been investigated for IVD cells.

This study aimed to investigate the potential of sulfated alginate 3D culture strategies, which have been applied to articular chondrocytes for IVD cells. The behavior of human NP cells was investigated using this novel sulfated alginate approach with varying degrees of sulfation (DS) to investigate the increasing sulfation effect in human NP cells.

## 2. Results and Discussion

### 2.1. NP Cell Density Significantly Decreases in 0.2 DS Sulfated Alginate Carriers

To explore the effects of sulfation in NP cells on cell growth and density, freshly isolated primary cells isolated from discs removed due to traumatic injury were encapsulated in standard alginate, 0.1% degree of sulfation (DS) and 0.2% DS alginate material at a density of 4 × 10^6^ cells/mL (1.08 × 10^5^ cell/ carrier) for up to 14 days under hypoxia (1.5% O_2_ tension). To quantify cell density after 7 and 14 days of culture, human NP cells inside the different carriers were stained with calcein-AM and ethidium homodimer for confocal slice-stack imaging and measured on the surface (Y orientation) and core of the carrier (X orientation) of each carrier (Figure 1a, Appendix A). For the analysis, the slice containing the highest total cell number from the slice stack was determined, and then, the number of slices between the first and last slices containing over half of this maximum cell number was examined. Hence, slices containing higher cell density were analyzed, providing more consistent slice stacks and avoiding edge effects and arbitrary areas. Imaging highlighted that 0.1% DS sulfated alginate and unmodified alginate carriers showed increased NP cell density after seven days of culture in both surface and core regions compared to the initial cell seeding amount (1.08 × 10^5^ cells per carrier), suggesting cellular proliferation inside the carriers (Appendix A). The 0.2% DS carriers exhibited a decreased NP cell density on the surface region compared to the initial cell seeding density after seven days of culture (Appendix A). In addition, all sulfated and unmodified alginate carriers exhibited a slight decrease in density during the second week of culture and no significant differences in cell density were observed between the surface and core regions in standard and 0.1% DS carriers after 7 or 14 days of culture (Appendix A). It is noteworthy that the core region measurements of the 0.2% DS carriers were not achieved due to the low stiffness of the carriers impairing core location after carrier halving. Interestingly, a significant decrease in cell density was observed in 0.2% DS sulfated carriers compared to standard alginate after 7 days and 14 days of culture (Figure 1b). Nevertheless, donor variability was higher in standard alginate than in sulfated groups (Appendix A). Additionally, 0.2% DS alginate carriers showed abnormal living cell clustering in the surface region after 14 days of culture, frequently observed in degenerative NP tissue samples (Figure 1c).

### 2.2. The 0.2% DS Sulfated Alginate Significantly Decreases NP Cell Viability after 7 Days 3D Culture

To further assess the influence of the different sulfated and non-sulfated materials on human NP cells, cell viability was assessed in previously measured slice stacks. Non-significant differences in cell viability were observed between standard and 0.1% DS groups in both the surface and the core of the carrier after seven days of culture (Figure 2a). Nevertheless, cell viability was significantly decreased (*p* < 0.05) in 0.2% DS sulfated carriers compared to 0.1% DS sulfated alginate in the surface of the carrier after seven days of culture (Figure 2a). In contrast, similar cell viability was evidenced between the core and surface regions in standard and 0.1% DS material. After 14 days of culture, standard alginate and 0.1% DS sulfated material exhibited higher living cell numbers, while 0.2% DS showed a significant reduction despite exhibiting similar percentage cell viability, suggesting that cells which had died during earlier time points had been degraded and were no longer detectable as dead cells, improving remaining cell viability percentage (Figure 2b). No significant differences in cell viability were observed between surface and core regions (Figure 2).

### 2.3. No Significant Differences in Cell Distribution Were Observed between Sulfated and Non-Sulfated Material at the Surface of the Carrier

To estimate the cell distribution in the Y-dimension near the surface of the non-sulfated and sulfated alginate carriers, the number of slices containing equal or over half of the total of the maximum number of cells was examined in each case. No significant differences were observed between the standard alginate, 0.1% and 0.2% DS sulfated alginate groups, indicating a similar cell distribution on the surface of the carriers on the Y-axis after 7 and 14 days of culture (Figure 3).

### 2.4. Anisotropic Cell Distribution Was Observed between Surface and Core of the Carrier in Standard Alginate Carriers

Slice-stack scans were performed in Y and X orientations to further investigate the cell distribution inside the carriers in different locations including surface and core. According to the number of slices containing over half of the maximum total cell number, no significant differences were observed between surface and core location (Y- and X-orientations) in standard alginate carriers after seven days of culture. Nevertheless, a significant increase in the slice number was observed in the core compared to the surface after 14 days of culture, indicating a more spread cell distribution between these locations as well as an anisotropic cell distribution depending on the scanning orientation (Figure 4a and Appendix A). Interestingly, 0.1% DS sulfated material did not exhibit any cell-related anisotropic behavior after 7 or 14 days of culture due to a similar slice number containing half of the maximum cell number in both surface (Y-orientation) and core (X-orientation) (Figure 4b).

### 2.5. Metabolic Activity and DNA Quantification

Additionally, human NP cell DNA and metabolic activity was measured to evaluate cell proliferation and performance. No significant differences in DNA content were observed between non-sulfated and sulfated carriers on the encapsulation day (day 0) or after 14 days. Notably, decreased DNA content in 0.2% DS sulfated alginate was found after 14 days of culture (Figure 5a). Additionally, sulfation-dependent metabolic activity reduction was evidenced between non-sulfated and sulfated carriers. In particular, 0.2% DS sulfated carriers showed a strong trend towards decreased metabolic activity on day 0 and after 14 days, suggesting a sulfation-dependent decrease in cell performance (Figure 5b).

### 2.6. Low RNA Yield on Sulfated Material Hampered Gene Expression Analysis

To evaluate the influence of sulfation in human NP cell phenotype recovery after cell expansion, genes considered markers for ECM synthesis (*ACAN*, *COL2A1*, *COL10A1*, *COL1A2*) as well as 18S as a reference gene were analyzed. According to 18S gene expression in the different carriers after seven days of culture, a sulfation-dependent increase in 18S Ct value was observed across the groups. Particularly, a significant increase in 18S Ct value was measured in the 0.2% DS group compared to standard alginate, indicating a higher cycle number was required to reach the fluorescence threshold in the qPCR method, a lower initial cDNA and thus lower yield in RNA extraction (Figure 6a). Unfortunately, the expression of anabolic markers was not observed in the sulfated material during the first 40 cycles which was attributed to the late 18S Ct values that were obtained. In contrast, *ACAN* and *COL2A1* gene expression increased within the standard alginate group after 7 and/or 14 days of culture, supporting the NP cell phenotype. In addition, *COL10A1* and *COL1A2* gene expression decreased after 14 days (Figure 6b, *p* = 0.1) suggesting a decreased hypertrophic phenotype and the return of a normal NP phenotype after the de-differentiation towards fibroblast-like cells which is induced in monolayer culture [46]. Overall, standard alginate 3D culture displayed an optimal NP cell phenotype and performance across the donors, while the cell phenotype evaluation was not achieved in sulfated carriers.

### 2.7. Discussion

The negatively charged ECM plays a vital role in cartilage and IVD tissues, promoting specific cues for resident cells and maintaining the tissue’s hydration. Over the past years, several hydrogels have been inspired in native ECM characteristics to provide novel 3D culture systems. In particular, sulfated alginate hydrogels have seen a growing interest as an improved 3D culture system for cartilage cell culture. In the present study, 0.1% and 0.2% DS alginate hydrogels were evaluated for IVD tissue engineering applications and were thus characterized according to their biological properties.

For that purpose, cell viability and cell density inside standard and sulfated alginate were monitored to investigate the impact of the sulfation degree in human NP cell culture. Previous studies in bovine and human articular chondrocytes encapsulated in unmodified and sulfated alginate in normoxia (21% O_2_) have reported a high chondrocyte viability in all sulfated or unmodified hydrogels [43,47,48] without significant differences between sulfated or non-sulfated materials. Similarly, both studies have evidenced sulfation degree-dependent cell morphological changes after three weeks, implying an extensive cell spreading throughout sulfated materials. A spread cell morphology indicates higher cellular recognition and cell adhesion, commonly obtained in hydrogels modified by integrin-binding motifs [49]. For instance, integrin β1 has been reported in sulfated alginate hydrogels as a chondrocyte-spreading mediator [47]. In contrast, in our study, we observed a significantly decreased human NP cell viability (*p* > 0.05) and density (*p* > 0.05) in the 0.2% DS carrier after seven days of culture. Additionally, NP cell proliferation was mainly observed in both surface and core regions of standard alginate carriers with an increased density value compared to the seeding density amount after seven days of culture. Interestingly, cell density decreased after 14 days of culture in all the groups together with no significant differences in cell viability, probably due to the large loss of cells after 7 days of culture. According to previously reported cell morphology changes in sulfated carriers, rounded-shaped NP cells were observed as sulfated alginate carriers, possibly due to the shorter culture period inferior to 3 weeks. Furthermore, abnormal cell clustering was also observed in 0.2% DS carriers after 14 days of culture, considered as a degenerative human NP tissue feature according to the standardized histopathology scoring system for human intervertebral disc degeneration [50].

A possible cause for observing different cell viability outcomes might have been the utilization of fetal calf serum (FCS) during the 3D culture period. FCS-containing media promotes significantly higher GAG production in human NP cells and a higher proliferation of bovine NP cell 3D cultures compared to serum-free media [51]. Similarly, previous investigations in bovine, equine and porcine articular cartilage chondrocytes encapsulated in alginate cultures containing FCS have exhibited improved cell performance with significantly higher proteoglycan synthesis compared to the serum-free alternative [52,53]. Interestingly, although no significant cell viability differences were reported under different FCS conditions, chondrocytes embedded in alginate and cultured in serum-free media did show lower (86 ± 1%) cell viability compared to 5% FCS containing cultures (97 ± 7%). This could have indicated an enhanced viability in the presence of serum. Furthermore, several growth factors are present in the undefined FCS cocktail that have been claimed as NP cell proliferation and survival factors including transforming growth factor β (TGFβ), platelet-derived growth factor (PDGF), fibroblast growth factor (FGF) and insulin-like growth factor (IGF) [54,55,56,57]. The presence of these mediators could explain chondrocyte long-term maintenance in previous studies, especially in sulfated alginate hydrogels, since they confer higher growth factor entrapment and participate actively in mediating the interaction between growth factors and their receptors [43]. Thus, the disparity between previous investigations containing FCS and our study could be explained by the presence of FCS and phenotypic differences between cell types. Noteworthy, composition and biological effect variability between FCS batches previously observed in chondrogenic differentiation of human adipose-derived MSCs (hAMSCs) [58], together with the poor clinical translation, are major concerns for FCS utilization. Additionally, culture media lacking ascorbic acid as well as different oxygen tension conditions has been used in our current study compared to previous sulfated alginate studies, suggesting another possible reason for result disparity [43,47,48].

To further investigate cell growth and distribution inside unmodified and sulfated cylindrical-shaped carriers, NP cells were stained for slice-stacks imagining in different locations including the surface and core of the carrier as well as orientations including Y and X orientations (Figure 1a). For the analysis, the slice-stack containing over half of the highest total cell number were only considered. Thus, the different number of slices conforming the slice-stack were examined and indicated different cell distribution profiles inside the carriers. No significant differences in the Y orientated number of slices measured in the surface were evidenced between the standard, 0.1% and 0.2% DS alginate carriers after 7 and 14 days of culture, indicating a similar cell distribution and spreading inside the different carriers. Nevertheless, a significantly higher number of slices containing over half of the total cell number (*p* > 0.05) were analyzed in X-orientated (core) measurements compared to Y-orientation (surface) in standard alginate constructs, suggesting different cell distribution depending on the location and orientation. Notably, this effect was not observed in 0.1% DS carriers (Figure 4b). Birefringence in alginate cylindrical gels was reported 70 years ago [59], identifying alginate as an anisotropic material depending on the cation flow gradient during gel casting. In this study, cylindrical-shaped carriers were cast, applying Ca^2+^ cations on the top and bottom of the carriers (Y orientation) creating a vertical Ca^2+^ gradient towards the core of the cylinder. In contrast, previous studies have utilized radial-oriented (X orientation) casting methods according to the QGel SA disc caster system [47,48], promoting a radial gradient of Ca^+2^ cations (Figure 7). In our study, different orientations were used to evaluate cell distribution. Interestingly, a significantly higher cell distribution was observed in standard alginate carriers measured in the X orientation, perpendicular to Ca^+2^ cation flow and parallel to the vertical Ca^+2^ cation gradient, indicating higher human NP cell distribution in planes with the same gel properties (Figure 7).

Nevertheless, not only orientations but also locations (surface and core) were different during these measurements, suggesting a combined effect of the orientation and location in cell distribution. Remarkably, surface and core locations in 3D microgel culture models have shown different cell performances and behaviors in the past. Previous studies in spherical microgels consisting of gelatin norbornene (GelNB) and a poly(ethylene glycol) (PEG) cross-linker have reported human chondroprogenitor cells, articular chondrocytes and bone marrow-derived mesenchymal stromal cells (hBMSCs) migration towards the surface, leaving the core region with less cells after seven days of culture [60,61]. One hypothesis to explain heterogeneous cell distribution inside the microgels is the higher growth factor concentrations and mass transfer within the boundaries of the hydrogel. Thus, encapsulated cells could sense and migrate away from the interior of the spheres, which has been also reported in previous investigations [62,63]. In addition, a sulfated version of gelatin/hyaluronic acid microgels has also observed a similar cell migration phenomenon from the core of the microgels to the surface after 14 days of culture. In this case, a nutritional gradient experienced by hBMSCs explains cell migration [47,48]. Nevertheless, parallel experiments with bulk gels (thickness ~3 mm) did not exhibit cell migration processes in line with our study, where we showed no significant differences in cell density between core and surface regions in standard and sulfated alginate carriers (4 mm ⌀ × 2 mm height) (Figure 1b). Thus, differences in cell distribution between the surface and core of standard alginate carriers might not have been directly affected by the measurement location but by the orientation of the measurements.

The analysis of DNA quantification revealed similar DNA amounts in all the groups on the encapsulation day but with a decreased DNA amount in 0.2% DS carriers after 14 days of culture (Figure 5a). Similarly, the analysis of cell performance according to cell metabolic activity revealed a sulfation-dependent reduction in mitochondrial activity on encapsulation day and after 14 days of culture. Particularly, human NP cells encapsulated in 0.2% DS carriers exhibited a trend (*p* = 0.051) towards decreased metabolic activity (*p* = 0.05) on the encapsulation day, and after 14 days, this strong trend (*p* = 0.07) towards decreased metabolic activity was maintained (Figure 5b). However, previous studies have shown the opposite effect, with higher and more homogeneous Col2 and proteoglycan deposition in sulfated alginate carriers containing human and bovine chondrocytes after 3 and 6 weeks, suggesting an enhanced chondrocyte cell performance. Furthermore, the DNA amount and chondrocyte proliferation were significantly higher in sulfated carriers, indicating a strikingly better cell performance in the presence of sulfated material [43,47,48]. Unfortunately, we did not observe similar results with human NP cells.

Finally, the gene expression of ECM genes, including *ACAN*, *COL2A1*, *COL1A2*, and *COL10A1*, were evaluated together with the *18S* ribosomal RNA reference gene. Predictably, the *18S* Ct value was significantly higher (*p* > 0.01) in the 0.2% DS groups, indicating a lower RNA amount and thus significantly decreased cell viability and density (*p* > 0.05) and a reduced DNA amount, as previously mentioned. Thus, the gene expression results of ECM markers were not achieved in the sulfated groups. Interestingly, previous research has shown no significant differences between sulfated and unmodified alginate carriers in *COL2*/*COL1* gene expression ratio in bovine chondrocytes [47]. Moreover, standard alginate cultures containing bovine chondrocytes have shown higher *COL2* expression at different time points (7, 14 and 21 days) compared to the sulfated material [43]. This could also be attributed to a higher stiffness in standard alginate since chondrocytes tended to express *COL2* in stiffer carriers [45]. Nevertheless, a previous study in human articular chondrocytes showed significantly decreased catabolic and pro-inflammatory markers, including IL-6, IL-8 and COX-2 in sulfated materials compared to the standard alginate, indicating a catabolic environment suppression in the presence of sulfated material rather than an anabolic environment enhancement [48]. In our study, “standard” alginate carriers showed an increased *COL2A1* and decreased *COL10A1* and *COL1A2* after 14 days of culture, although longer 3D culture periods have been reported to recover ACAN expression. Taken together, standard alginate seems to be the most suitable 3D culture model for human NP cell culture. Nevertheless, limited RNA extraction from sulfated material has left the sulfation effect in the human NP phenotype obscured. Hence, further investigations of catabolic and anabolic makers in NP cells encapsulated in sulfated alginate 3D cultures are needed to clarify the sulfated alginate role in IVD cell culture.

## 3. Conclusions

In conclusion, human NP cells encapsulated in “gold standard” alginate have shown significantly increased cell viability and metabolic activity and enhanced density and DNA amount compared to 0.2% DS sulfated alginate alternative. Additionally, standard alginate material seems to provide different cell distribution depending on the orientation and location in the carrier. This outcome might be explained as a cell preference to growth or spread depending on the location and calcium gradient orientation during carrier casting. Hence, future work employing cLSM slice-stack imaging on hydrogels might consider different orientation preferences according to the hydrogel casting system as well as different location alternatives. According to gene expression data, standard alginate showed optimal cell phenotypic support. Unfortunately, anabolic and catabolic gene expression was not obtained in the sulfated material due to a low RNA yield. Taken together, a standard alginate 3D culture model is the preferred option for human NP cell culture. Nevertheless, a further examination of sulfation degree impact on human NP cell phenotype is needed to characterize sulfated alginate biological properties on IVD culture.

## 4. Materials and Methods

### 4.1. Ethical Approval

Human IVD tissue was obtained from patients undergoing spine surgery for the treatment of trauma with the informed consent of the patients or relatives. This study utilized six surgical samples from six different individuals.

### 4.2. Human NP Cell Isolation

Human IVD tissue was collected from patients (Table 1) aged between 30 and 72 years old (51.0 ± 15.7 [mean ± SD]) and subsequently separated into NP, AF and CEP tissue for processing within 24 h after the surgery. Human NP cells were isolated from NP tissue pieces by 1 h digestion with 1.9 mg/mL pronase ((7 U/mg) #10165921001; Roche Diagnostics, Basel, Switzerland) followed by overnight digestion with collagenase II ((285 U/mg) Worthington, London, UK) in serum free low-glucose (1g/L) Dulbecco’s Modified Eagle Medium (LG-DMEM; 10569010 HG; Gibco, Zug, Switzerland) containing 1% *v*/*v* P/S and 0.2% *v*/*v* primocin (#ant-pm; InvivoGen, San Diego, CA, USA, distributed by Lubioscience, inc., Lucerne, Switzerland) on an orbital shaker at 37 °C. Following digestion, the remaining tissue was removed using 100 μm cell strainer filtration and cell viability and counting were determined via trypan blue exclusion. Following isolation, human NP cells were expanded until passage three in monolayer in HG DMEM supplemented with 10% *v*/*v* heat-inactivated FCS (#F7524; Sigma-Aldrich, Buchs, Switzerland), 1% *v*/*v* P/S, 2.5 μg/mL Amphotericin B (#A4888; Sigma-Aldrich) and maintained at 37 °C in a humidified atmosphere containing 5% CO_2_ and 21% O_2_ prior to encapsulation.

### 4.3. Cell Encapsulation

Standard alginate, 0.1% *v*/*w* degree of sulfation (DS) and 0.2% DS alginate material were synthesized as described previously [43]. In short, primary human NP cells were expanded until passage three. Afterwards, the solutions of 2.5% of standard alginate, 0.1% DS and 0.2% DS alginate were mixed with human NP cells at 4 × 10^6^ cells/mL physiological density [64] and 27 µL of each mixture were pipetted onto cylindrical-shaped silicon ring molds of 6 mm in outer and 4 mm in inner diameters and 2 mm in height. Silicon molds containing standard alginate, 0.1% DS and 0.2% DS alginate solutions were immersed in 0.2 M CaCl_2_ for 10 min at 37 °C. Disc-shaped carriers were then washed twice with 0.15 M NaCl and low-glucose (1g/L) Dulbecco’s Modified Eagle Medium (LG-DMEM; cat.# 10569010 HG; Gibco) containing 1% *v*/*v* P/S. Finally, disc-shaped carriers were cultured for two weeks for phenotype recovery in serum-free low glucose (1g/L) DMEM under NP tissue lowest physiological hypoxia condition (1.5% O_2_ tension) and re-crosslinking every week with 0.2 M CaCl_2_ at 37 °C for 10 min. All experimental groups were maintained in an Invivo II 300 Hypoxia Workstation (Ruskinn Technology, Ltd., Bridgend, UK) throughout the culture. Finally, carriers were collected on day 0, 7 and 14 for the downstream analysis of cell viability, density, DNA quantification and metabolic activity.

### 4.4. Cell Density and Viability

To assess the cell density and viability of NP cells in each alginate material, disc-shaped carriers were cultured in serum-free LG medium containing 5 µM calcein-AM (#17783-1MG; Sigma-Aldrich) to stain the living cells and 1 µM ethidium homodimer (#46043-1MG-F; Sigma-Aldrich) to stain the dead cells and incubated at 37 °C for 90 min. Following the incubation and after washing carriers with phosphate-buffered salt solution (PBS), 3D stacked images were taken on a confocal laser scanning microscope (cLSM710; Carl Zeiss; Jena, Germany). To facilitate and compare the analysis of human NP cell location, cylinder-shaped carriers were measured at the surface region at a Y-orientation and core region at an X-orientation. Afterwards, the maximum number of live and dead cells was determined in each slice-stack measurement. For the analysis, slices containing over half of the maximum total cell number were examined using a custom-made macro for ImageJ software Java 1.8.0_172 version (see also [65]). For density quantification, a correction factor was applied to avoid cell number overestimation [66], (Appendix A).

### 4.5. DNA Quantification

All samples were digested overnight in 5 mM L-cysteine hydrochloride (#20119; Sigma-Aldrich) enriched papain solution (3.9 U/mL; #P3125; Sigma-Aldrich) at 60 °C. The DNA content was measured from digested alginate carriers using Hoechst 33,258 dye (#86d1405; Sigma-Aldrich). The optical fluorescence index was measured at 350 nm excitation and 450 nm emission wavelengths in digested samples as well as in a standard curve of increasing DNA sodium salt concentrations from calf thymus (#D1501; Sigma-Aldrich) to interpolate the sample DNA concentration.

### 4.6. Metabolic Activity

For metabolic activity evaluation, on day 0 and 14 standard, 0.1% and 0.2% alginate carriers containing NP cells were immersed in HG-DMEM supplemented 10% FCS (Sigma–Aldrich) containing 50 μM resazurin sodium salt solution and incubated for two hours at 37 °C. The optical fluorescence index was measured at 544 nm excitation wavelength and 578 nm emission wavelength using an ELISA plate reader (Spectramax M5, Molecular Devices, distributed by Bucher Biotec, Basel, Switzerland).

### 4.7. RNA Extraction, cDNA Synthesis and Relative Gene Expression

Sulfated and non-sulfated carriers were collected, snap-frozen in liquid nitrogen and pulverized using a cooled mortar. The powder was then mixed with Tri reagent and processed as described previously using a combined Trizol-silicon membrane purification approach [67,68].

In a next step, the isolated RNA (9–22 ng/µL) was transcribed into cDNA using a High-Capacity cDNA Reverse Transcription kit (#4368814; Thermo Fisher Scientific, Waltham, MA, USA) and a MyCycler™ Thermal Cycler system (#1709703; Bio-Rad Laboratories, Inc.; Cressier, Switzerland).

Finally, the obtained cDNA was then mixed with the primers of interest (Table 2) and iTaq Universal SYBR Green Supermix (#1725122; Bio-Rad) for a quantitative polymerase chain (qPCR) reaction on a CFX96™ Real-Time System (#185-5096; Bio-Rad Laboratories). Finally, the 2^−ΔΔCt^ method [69] was used for relative gene expression analysis with 18S ribosomal RNA as the reference.

### 4.8. Statistical Analysis

A nonparametric distribution was assumed in all quantitative data. Cell density, viability, DNA quantification and metabolic activity data were analyzed using a Kruskal–Wallis test followed by a Dunn’s multiple comparisons test using GraphPad Prism (version 10.2.1 for Mac OS X, GraphPad Software; San Diego, CA, USA) and a *p*-value < 0.05 was considered statistically significant. All quantitative results are presented as median and up to six replicates were used for each experiment. However, the exact number of biological replicates (n) are indicated in each figure legend.

## Figures and Tables

**Figure 1 gels-10-00330-f001:**
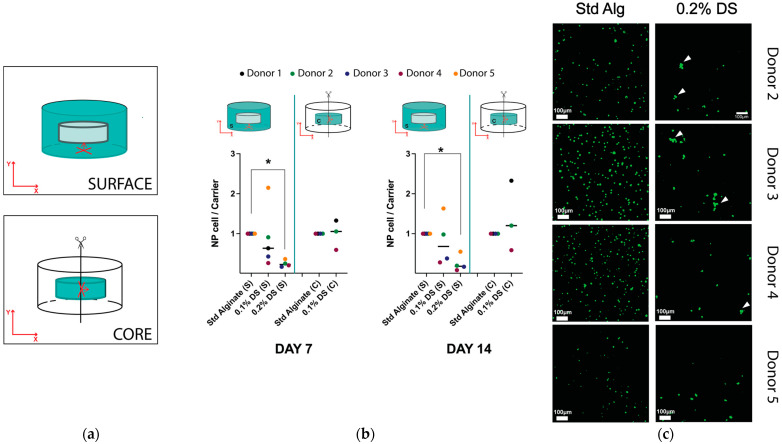
(**a**) Scheme of the orientation and imaging strategy of the cylinder carriers during cLSM slice-stack imaging on the surface or in the core. (**b**) Human NP cell density in standard, 0.1% DS and 0.2% DS alginate carriers normalized to standard alginate group after 7 and 14 days of culture. (**c**) Calcein-AM staining on human NP cells encapsulated in standard and 0.2% DS alginate carriers after 14 days of cell culture. White arrows indicate human NP living cell clustering. Shown are medians, N = 3–5, *p*-value: * <0.05. Objective magnification = 10×; scale bar = 100 µm. Abbreviations: C, core; S, surface. Red arrows indicate Y and X axis.

**Figure 2 gels-10-00330-f002:**
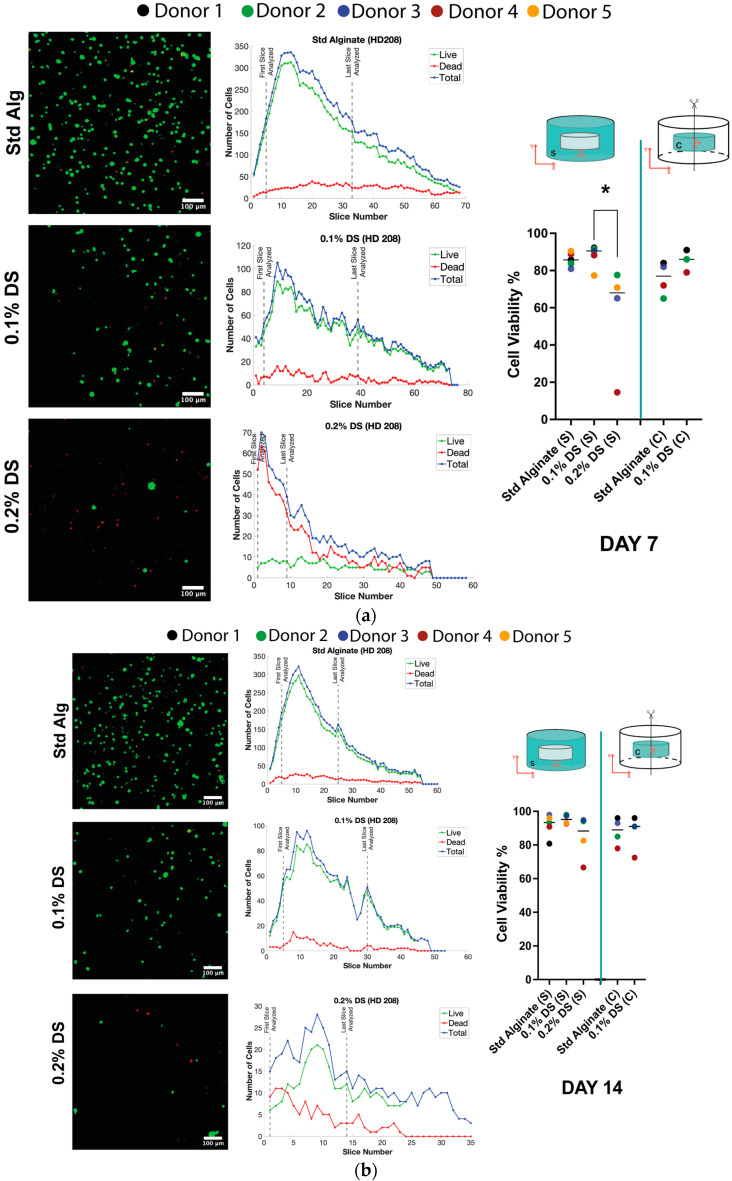
Live (green) and dead (red) staining in human NP cells in the surface or core of standard, 0.1% DS and 0.2% DS alginate carriers: (**a**) Cell viability after 7 days of 3D culture; (**b**) cell viability after 14 days of 3D culture. Shown are medians ± standard deviation, N = 3–5, *p*-value: * <0.05. Objective magnification = 10×.; scale bar = 100 µm. Abbreviations: C, core; S, surface. Red arrows indicate Y and X axis.

**Figure 3 gels-10-00330-f003:**
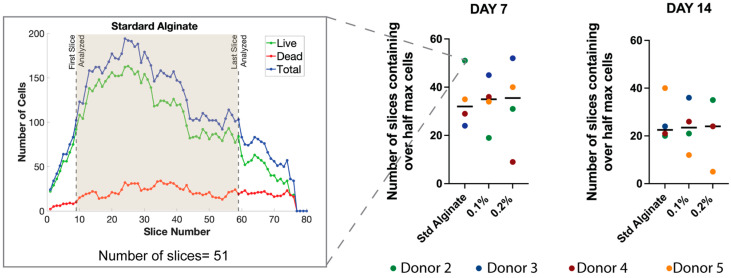
Comparison of the number of 2D slices obtained using confocal laser scanning microscopy (cLSM) containing over half the maximum total cells number between non-sulfated and sulfated carriers. Shown are medians, N = 4.

**Figure 4 gels-10-00330-f004:**
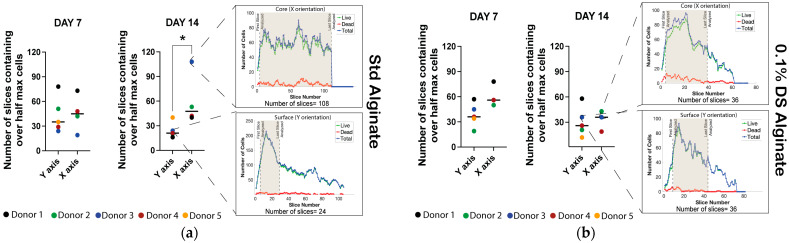
Human NP cell distribution in the surface and core of standard and 0.1% DS alginate carriers. (**a**) Y and X oriented number of slice-stacks containing over half of the maximum cell number on the surface and core of standard alginate carriers after 7 and 14 days of 3D culture. (**b**) Y and X oriented number of slice-stacks containing over half of the maximum cell number on surface and core 0.1% DS sulfated alginate carriers after 7 and 14 days of 3D culture. Shown are medians, N = 3–5, *p*-value: * <0.05.

**Figure 5 gels-10-00330-f005:**
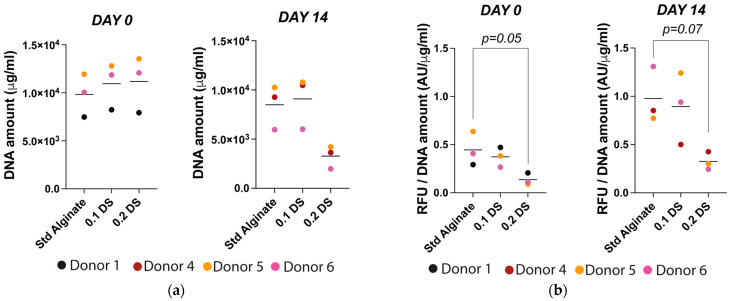
(**a**) Human NP cell DNA content on day 0 and day 14 in standard alginate, 0.1% and 0.2% DS sulfated alginate. (**b**) Mitochondrial activity in standard alginate, 0.1% and 0.2% DS sulfated alginate on day 0 and day 14. Shown are medians, N = 3–5, *p*-value: * <0.05.

**Figure 6 gels-10-00330-f006:**
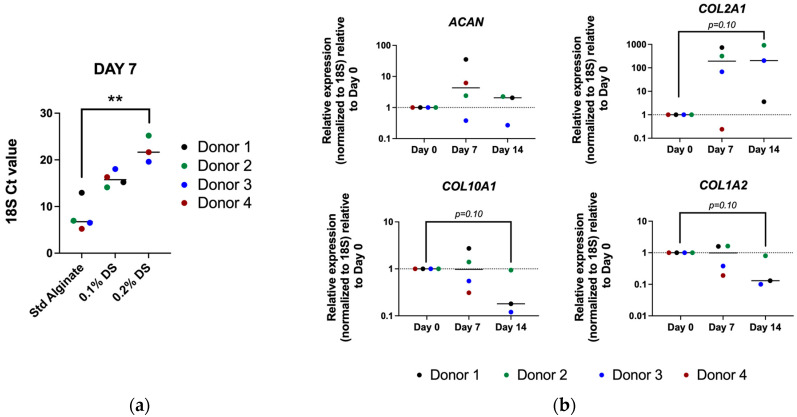
(**a**) 18S reference gene C*t* values quantified in standard alginate, and in 0.1% and 0.2% DS sulfated alginate, respectively, after 7 days of culture. Shown are the medians, N = 3–4, *p*-value: *p*-value ** <0.01; (**b**) Relative gene expression of NP-specific ECM markers, i.e., *ACAN*, *COL2A1*, *COL10A1* and *COL1A2* in human NP cells encapsulated on standard alginate after 7 and 14 days of culture. Shown are medians, N = 3–4.

**Figure 7 gels-10-00330-f007:**
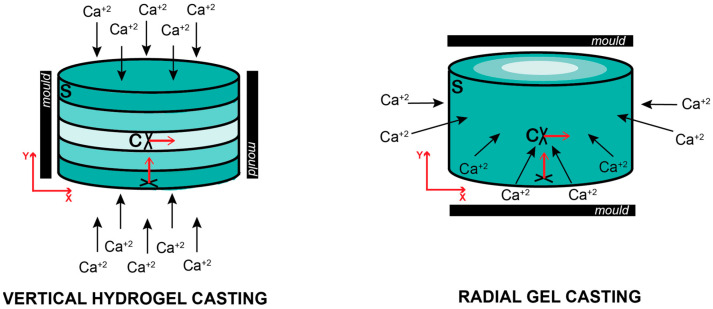
Vertical (**left**) and radial (**right**) alginate hydrogel casting represented with Ca^+2^ gradient. Red arrows indicate Y and X axis.

**Table 1 gels-10-00330-t001:** Patient donor list of NP tissue samples enrolled in the study.

Donor ID *	Level *	Age	Sex	Surgery Reason
Donor 1	Th12/L1	30	male	Trauma
Donor 2	L2/L3	34	male	Trauma
Donor 3	Th12/L1	72	female	Trauma
Donor 4	L1/L3	34	male	Trauma
Donor 5	L4/L5	48	female	Trauma
Donor 6	L1/L2	37	female	Trauma

* Posttraumatic intervertebral disc subjects without a history of disc degeneration were collected and identified with their ID number. Abbreviations: L, lumbar; Th, thoracic.

**Table 2 gels-10-00330-t002:** Investigated gene overview and primers used in this study.

Gene Type	Full Name	Symbol	NCBI Gene ID	Forward and Reverse Primer Sequences
Reference Gene	18S ribosomal RNA	*18S*	100008588	f—CGG ACA GGA TTG ACA GAT TGA TAG r—TGC CAG AGT CTC GTT CGT TA
NP cell ECM markers	Aggrecan	*ACAN*	176	f—CAT CAC TGC AGC TGT CAC r—AGC AGC ACT ACC TCC TTC
	Collagen type 2, Alpha chain 1	*COL2A1*	1280	f—AGC AGC AAG AGC AAG GAG AA r—GTA GGA AGG TCA TCT GGA
	Collagen type 10, Alpha chain 1	*COL10A1*	1300	f—GAA TGC CTG TGT CTG CTT r—TCA TAA TGC TGT TGC CTG TTA
	Collagen type 1, Alpha chain 2	*COL1A2*	1278	f—GTG GCA GTG ATG GAA GTG r— CAC CAG TAA GGC CGT TTG

## Data Availability

The original data can be obtained upon request from the corresponding author.

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
