# Peer review of "Sulfated Hydrogels as Primary Intervertebral Disc Cell Culture Systems"

_gels, 2024, doi:10.3390/gels10050330_

Round 1
Reviewer 1 Report
Comments and Suggestions for Authors
The work described in the manuscript was performed at a high quality level. There are some typos in the article (see attached PDF file), which do not in any way affect the overall assessment of the study described.

Author Response
Reviewer 1
The work described in the manuscript was performed at a high-quality level. There are some typos in the article (see attached PDF file), which do not in any way affect the overall assessment of the study described.
Answer: The authors thank the reviewer for their feedback and their recognition of the value of this manuscript. The typos were corrected according to the reviewer’s comments and some subchapters titles were shortened as kindly requested.
Reviewer 2 Report
Comments and Suggestions for Authors
Thank you for the opportunity to review this interesting manuscript. On my opinion, the article is suitable for this particular journal and brings new knowledge within its field.
The abstract and introduction are well written and comprehensive. The introduction brings a good insight and start to the article content. Here, I suggest to correct in line 60, 61 the statement: ‘IVDs are located between two adjacent vertebral bodies and are responsible for spinal motion … ' The IVD is not responsible for spinal motion. Pleas rewrite.
The results are well written and concise. However, can authors explain here, how they coped with isolated NP cells, since they are prone to dedifferentiation? Did they use a primary culture? How many days did the cells grow in culture? Any changes observed in the culture? These cells usually dedifferentiate after 1st or 2nd passage.
The methods and discussion are well written and I do not have any special suggestions. The conclusion is ok.
I recommend minor revision. The article is otherwise well written and suitable for publication.
Author Response
Thank you for the opportunity to review this interesting manuscript. On my opinion, the article is suitable for this particular journal and brings new knowledge within its field.
Answer: The authors thank the reviewer for their feedback and their recognition of the value of this manuscript.
The abstract and introduction are well written and comprehensive. The introduction brings a good insight and start to the article content. Here, I suggest to correct in line 60, 61 the statement: ‘IVDs are located between two adjacent vertebral bodies and are responsible for spinal motion … ' The IVD is not responsible for spinal motion. Pleas rewrite.
Answer: We corrected the indicated statement and rephrase it as: IVDs are located between two adjacent vertebral bodies and contribute to weight bearing of the spine, withstanding compression loads, and providing spine flexibility (line 60, 61).
The results are well written and concise. However, can authors explain here, how they coped with isolated NP cells, since they are prone to dedifferentiation? Did they use a primary culture? How many days did the cells grow in culture? Any changes observed in the culture? These cells usually dedifferentiate after 1st or 2nd passage.
Answer: The authors thank the reviewer for their comments. One of the aims in this review was indeed to address human NP cell phenotype recovery in sulfated materials and compare it with standard alginate culture system. For that purpose, primary human NP cells were isolated and expanded until passage three (which is still considered low passage for these cells) and then encapsulated in the sulfated/non-sulfated materials.
After cell expansion, these cells usually need a phenotype recovery phase of 14 days in 3D environment. Thus, we aimed to evaluate the gene expression of phenotypic makers after 7 and 14 days of 3D culture and investigate the sulfation effect on cell re-differentiation. Unfortunately, phenotypic makers were not studied in the sulfated groups due to low cell number after 7 and 14 days of 3D culture (Figure 1, Figure 2 and Figure 6a). However, standard alginate exhibited increased NP phenotypic markers ACAN, COL2A1 gene expression (Figure 6b) while COL10A1 and COL1A2 gene expression decreased after 14 days (Figure 6b), suggesting a decreased hypertrophic phenotype and reversal of de-differentiation (i.e. return towards a more normal NP phenotype), ([46] Rutges, JP et al (2010) doi: 10.1016/j.joca.2010.08.006). We have improved subchapter 2.6. in order to clarify it.
The methods and discussion are well written and I do not have any special suggestions. The conclusion is ok.
I recommend minor revision. The article is otherwise well written and suitable for publication.
Answer: The authors thank the reviewer for their feedback and suggestions.

Reviewer 3 Report
Comments and Suggestions for Authors
Dear Authors,
The manuscript entitled "Sulfated hydrogels as primary intervertebral disc cell culture 2 systems" represents a very important study in the field and will be very valuable for the readers. The entire manuscript has been well prepared, and the methological approaches, described in the submitted manuscript are in detail performed. Indeed, the authors have performed a very good work and also the figures are presented, scale bars are included, while the same has been applied also for the statistics. I have only minor comments.
1) Please include, in the figure legend before, the scale bar, the original magnification of the images.
2) Do the authors think that biomechanical analysis or any other method for establishing mechanical properties or viscoelasticity can be applied to the hydrogels.
Thank you.
Author Response
Dear Authors,
The manuscript entitled "Sulfated hydrogels as primary intervertebral disc cell culture 2 systems" represents a very important study in the field and will be very valuable for the readers. The entire manuscript has been well prepared, and the methological approaches, described in the submitted manuscript are in detail performed. Indeed, the authors have performed a very good work and also the figures are presented, scale bars are included, while the same has been applied also for the statistics.
Answer: The authors thank the reviewer for their feedback and their recognition of the value of this manuscript.
I have only minor comments.
1) Please include, in the figure legend before, the scale bar, the original magnification of the images.
Answer: We have included the original magnification before the scale bars on Figure 1 and Figure 2.
2) Do the authors think that biomechanical analysis or any other method for establishing mechanical properties or viscoelasticity can be applied to the hydrogels.
Answer: The authors appreciate this suggestion. The biomechanical testing and rheological analysis including the chemical characterization of modified alginates and sweling properties of these hydrogels has been previously reported in [43] Öztürk E. et al (2016) doi:10.1002/adfm.201600092.
